# Trends in Airborne Chrysotile Asbestos Fibre Concentrations in Asbestos Cement Manufacturing Factories in Zimbabwe from 1996 to 2016

**DOI:** 10.3390/ijerph182010755

**Published:** 2021-10-13

**Authors:** Benjamin Mutetwa, Dingani Moyo, Derk Brouwer

**Affiliations:** 1Faculty of Health Sciences, School of Public Health, University of the Witwatersrand, Johannesburg 2193, South Africa; moyod@iwayafrica.co.zw (D.M.) derk.brouwer@wits.ac.za (D.B.); 2Faculty of Medicine and Health Sciences, Midland State University, Gweru 054, Zimbabwe; 3Department of Community Medicine, Faculty of Medicine, National University of Science and Technology, Bulawayo 029, Zimbabwe

**Keywords:** exposure, chrysotile asbestos, trends, personal exposure, airborne asbestos fibre concentration

## Abstract

Zimbabwe has two major factories that have been manufacturing chrysotile asbestos cement products since the 1940s. Exposure monitoring of airborne fibres has been ongoing since the early 1990s. This study examines trends in personal exposure chrysotile asbestos fibre concentrations for the period 1996–2016. Close to 3000 historical personal exposure measurements extracted from paper records in the two factories were analysed for trends in exposure. Exposure over time was characterised according to three time periods and calendar years. Mean personal exposure chrysotile asbestos fibre concentrations generally showed a downward trend over the years in both factories. Exposure data showed that over the observed period 57% and 50% of mean personal exposure chrysotile asbestos fibre concentrations in the Harare and Bulawayo factories, respectively, were above the OEL, with overexposure being exhibited before 2008. Overall, personal exposure asbestos fibre concentrations in the factories dropped from 0.15 f/mL in 1996 to 0.05–0.06 f/mL in 2016—a decrease of 60–67%. These results can be used in future epidemiological studies, and in predicting the occurrence of asbestos-related diseases in Zimbabwe.

## 1. Introduction

Asbestos is a generic term for a group of naturally occurring silicates that principally include the serpentine variety (white chrysotile asbestos) and the amphibole variety, consisting of crocidolite (blue asbestos) and amosite (brown asbestos) [1]. Asbestos exposure has drawn much international, regional, and national attention, as it presents significant public and occupational health concerns. All asbestos types are known to cause asbestos-related disease [1,2,3].

The World Health Organization reports that 125 million people worldwide are exposed to asbestos at the workplace, with 107,000 people succumbing to asbestos-related diseases annually [2]. Although amphibole production has all but ceased worldwide, chrysotile asbestos continues to be produced and used in some countries. While the production and use of asbestos in most developed countries has declined in recent years due to health concerns, and the subsequent ban of asbestos-containing products, there continues to be extensive production, sale, and use of chrysotile in South and Central America, Asia, and Africa [2,4]. Russia is the world’s leading producer of chrysotile asbestos; others include China, Kazakhstan, Brazil, and India, with production at Zimbabwe’s chrysotile mines stalling in 2010 due to economic challenges. Currently, there are efforts to resuscitate the mining of chrysotile asbestos, with tailings dumps being harnessed to extract fibres for the two chrysotile asbestos cement manufacturing factories in the cities of Harare and Bulawayo.

Zimbabwe has long been one of Africa’s major producers of chrysotile asbestos [5,6]. During the 1970s, production averaged 200,000 metric tonnes per annum, rising to a peak of 259,000 tonnes in 1979. However, production declined to 100,000 tonnes per annum for the period 2004–2007, and reduced drastically during the hyperinflation period of 2008 such that, by 2010, only 2400 tonnes were reported to have been produced [5]. Important chrysotile products that are produced in Zimbabwe include reinforced chrysotile asbestos roofing sheets and tiles, water pipes, heat-resistant or fire-resistant insulation materials, and packings and gaskets in the vehicles industry. The two chrysotile asbestos mines—the Shabanie and Mashava mines—had a combined production capacity of 140,000 metric tonnes of chrysotile asbestos in the 1980s and 1990s; 90% of this product was exported, with 10% consumed by the local chrysotile asbestos cement manufacturing industry [7].

From the early 1990s, ~7000 workers were engaged in mining and milling at the two major mines, with ~4000 engaged in the manufacturing of chrysotile asbestos products [8]. During the same period, it was reported that 40,000–45,000 people lived within a few kilometres of the mills and mines, and a large proportion of the population lived and worked in buildings with chrysotile asbestos [8]. Zimbabwe has two major factories that manufacture chrysotile asbestos products, and which have been the main users of chrysotile asbestos since their establishment in the 1940s and 1950s.

A limited number of papers have reported temporal trends in personal exposure chrysotile asbestos fibre concentrations in chrysotile asbestos cement manufacturing plants. In Germany, it was observed that there was a decrease in asbestos dust concentrations for the period 1950 to 1990, which was attributed to the rapid decline in the use of asbestos since 1980, when regulations and bans on the production, use, and placement of asbestos on the market were introduced [9,10]. Furthermore, Coble et al. reported that there was a 5% decline in asbestos exposure observed during compliance inspections of pulp and paper facilities [11]. In another study of exposure–response relationships for asbestos-related diseases, Finkelstein reported declines in exposure for the years 1949, 1969, and 1979, with estimates recorded as 40 f/mL, 20 f/mL, and 0.2 f/mL for willow operators, 16 f/mL, 8 f/mL, and 0.5 f/mL for forming machine operators, and 8 f/mL, 4 f/mL, and 0.3 f/mL for lathe operators for 1949, 1969, and 1979, respectively [12]. Additionally, declines in asbestos exposure were reported in asbestos cement plants in Sweden [13], South Africa [14], Japan [15], Yugoslavia, Poland, and Latvia [16], and the USA [17]. These studies show that exposure during the earlier years was high—particularly during the 1970s—compared to the period 1990 to 2000s.

To control and reduce exposure, various regulatory agencies dealing with occupational safety and health (OSH) have established occupational exposure limits (OELs) or threshold limit values (TLVs) for airborne asbestos fibres. It is expected that workers exposed repeatedly to levels at or below the OEL are at low risk of developing adverse health effects [18,19].

OELs have been declining over the years in response to new information on exposure–response effects experienced by workers and/or experimental animals, thereby influencing the exposure levels observed in various asbestos workplace settings. In the USA, OELs have moved from as high as 12 f/mL in the early 1970s, to 2 f/mL in the mid-1970s, 0.2 f/mL in the mid-1980s and, from 1995 to date, the OEL has been set at 0.1 f/mL. Today, most countries have aligned their asbestos OEL to that of the USA’s OSHA or ACGIH, which has been set at 0.1 f/mL [18,20].

In Zimbabwe, there is no specific legislative instrument that governs the management and enforcement of an OEL for chrysotile asbestos. Management is generally through non-specific regulations, such as the Statutory Instrument 68 of 1990 on Accident Prevention and Workers Compensation [21]. Moreover, there is no statutory OEL for chrysotile asbestos fibres, save for guidelines on OELs published by the National Social Security Authority (NSSA)—Occupational Safety and Health Division, which has set the limit at 0.1 f/mL for all forms of asbestos fibres [22]. Hence, the current OEL for chrysotile asbestos fibres in Zimbabwe is 0.1 f/mL. However, this OEL is not a statutory limit but, rather, a recommended limit, which is expected to be part of an envisaged asbestos regulation. In the absence of specific guidance on the management of chrysotile asbestos exposure, the chrysotile asbestos cement manufacturing industry has developed its own occupational exposure monitoring programme, where personal and static exposure sampling data have been collected since the 1980s, and more structured in the 1990s to the mid-2000s through to 2016. These data provide an opportunity to understand the extent of exposure to asbestos fibres in the Zimbabwean chrysotile asbestos cement manufacturing industry over the years.

## 2. Materials and Methods

### 2.1. Study Design

This secondary data analysis study was carried out in the two chrysotile asbestos cement manufacturing (ACM) factories situated in Harare and Bulawayo. The original data of personal exposure chrysotile asbestos fibre measurements were provided from the company factories to the authors following the company agreeing to access the records of the personal exposure measurements data. The data were extracted from the paper records of personal chrysotile asbestos fibre exposure measurements taken in the factories by company personnel. Data recorded from 1996 to 2016 by the two chrysotile asbestos cement factories were analysed for trends.

### 2.2. Collection of Measurements

Operational areas for which personal exposure data were available were cutting saws, fettling table, kollergang, moulded goods, ground hard waste, laundry room, sheeting planter mixer, lathe machining of pipes, and multi-cutter operations (Table 1). Generally, exposure data were collected once every month, though in some years, measurements depended on the availability of plant operations, sampling equipment, and consumables. Appendix A shows the number of measurements collected in the various operational areas.

### 2.3. Method of Chrysotile Asbestos Fibre Measurements

The written asbestos method on file in the factories showed that measurements of airborne asbestos fibre concentrations followed the standard method of the Asbestos International Association (AIA) Recommended Technical Membrane Filter Reference Method (AIA, 1982) [23]. As part of adherence to the AIA technical reference method, field blank filter samples were reported and used as controls as part of the quality control programme. In summary, a personal sampling pump set at a 1 L/min flowrate was connected to a sampling train, consisting of plastic tubing and a sample holder (cowl) with a 25 mm membrane filter. The whole sampling train of the pump, tubing, sample holder, and filter was hooked to a worker. The pump was then switched on, and sampling took place over a period of around four hours, after which the filters were removed, placed at the appropriate labelled slides, and treated with acetone vapour to clear. Using a hypodermic syringe, a drop of triacetin was placed onto the acetone-cleared filters and covered with a cover slip. The treated filters on the slides were stored for 24 h, after which counting of the fibres took place using a phase-contrast microscope. The limit of detection (LoD) for the method was 0.02 f/mL.

The period of 1996 to 2016 was divided into three time periods: 1996–2000, 2001–2008, and 2009–2016. During 1996–2000, the chrysotile asbestos cement manufacturing industry was in a self-regulatory mode with respect to safety and health standards, and the ACM manufacturing industry had an active exposure monitoring programme in light of the call to phase out the use of chrysotile asbestos. During these early years of the 1990s, the asbestos industry set its own chrysotile exposure limit of 0.2 f/mL and an action limit of 0.15 f/mL in the absence of a national statutory exposure limit on chrysotile asbestos. From 2001 to 2008, the chrysotile exposure monitoring program continued; however, there was a sharp decline in economic activity nationally. Monitoring of exposure continued for the period 2009 to 2016 against the backdrop of improved retooling of the industry and change from the use of locally produced asbestos to largely imported fibre.

### 2.4. Quality Assurance and Reliability of the Chrysotile Asbestos Fibre Exposure Data

From the early 1990s to 2011, correspondences at the two factories showed that the factories participated in an inter-laboratory quality assurance and control fibre-counting programme, which involved laboratories at the two chrysotile mines in Zimbabwe, another chrysotile asbestos cement plant laboratory in Zambia, the Department of Minerals and Energy in South Africa, and a French laboratory in Paris, with a view to improving the quality and reliability of exposure measurements. Additionally, as part of an oversight programme on quality control, in 2008, the Institute of Occupational Medicine (IOM), UK, was invited to conduct an independent evaluation of levels of chrysotile asbestos fibres in the ambient air around various work processes [24]. The independent evaluation of levels of chrysotile asbestos fibres in the two factories provided a good measure of reliability and assurance to the personal exposure chrysotile fibre concentrations generated by the company over the years, and subsequently used in the study described in this paper. The IOM reported that personal and static samplers were being correctly mounted on the workers, with proper positioning of sample holders in the workers’ breathing zones. They further reported that the company’s analytical laboratory was adequately equipped for the collection and measurement of airborne chrysotile asbestos fibres, and that there was good consistency between the IOM and the company’s calibration equipment for calibration flows of the sampling pumps [24]. These efforts demonstrated that the data used in this study provided a measure of reliability of the exposure values obtained in the factories over the years.

### 2.5. Data Description and Classification of Measurements

Approximately 3000 personal exposure measurements were collected in the operational areas (Appendix A) over the 21-year period in the two factories. Personal sampling points were classified into six production areas for both the Harare and Bulawayo factories; a further subclassification was made for the pipe section of the Bulawayo factory. For the two main ACM factories, personal sampling data were classified as described in Table 1. However, for laundry room (28 values) and sheeting plant mixer operations (30 values), actual measured values for the Bulawayo factory for the period 1996 to 2016 were too few and, thus, were not considered in the analysis. Additionally, for the pipe section, personal sampling data were classified into three broad areas—namely, (a) pipe plant operations—lathe machining asbestos pipe joints, (b) pipe plant—lathe machining of full-length asbestos sewer and water pipes, and (c) multi-cutter, where cutting of full-length pipes into collars for coupling of pipes was carried out. For the cutting saw operations, measurements were taken at ~4–6 saws per month. Personal sampling data for each broad operational area were averaged for each month (Appendix A).

These tasks were considered to have the highest potential for exposure to airborne chrysotile asbestos.

### 2.6. Statistical Analysis

Data analysis was conducted using IBM SPSS version 26. For analysis, monthly averaged personal exposure levels for the factories were used. Mean personal airborne chrysotile fibre concentrations were analysed per operational area per factory, and trends in airborne fibre concentrations over the years were displayed graphically.

### 2.7. Ethics

This study was approved by the University of the Witwatersrand Human Research Ethics Committee (clearance certificate number M181157) and the Medical Research Council of Zimbabwe (MRCZ) (approval number MRCZ/A/2445).

## 3. Results

There were 2890 personal samples collected over the 21-year period in the different operational areas of the two chrysotile cement manufacturing factories (Appendix A), and 1663 monthly averaged personal chrysotile asbestos fibre concentrations (Appendix A). The Harare factory had the greater proportion (63.9%) of monthly averaged concentrations. Table 2 and Table 3 show the summary statistics of the personal chrysotile asbestos fibre concentrations for the Harare and Bulawayo factories, respectively. Small variations in airborne chrysotile asbestos fibre concentrations were recorded at the fettling table operations (SD ± 0.02) for the Harare factory. The other operational areas in both factories showed variation in airborne chrysotile asbestos fibre concentrations, ranging from 0.01 to 0.30 f/mL. Fettling table operations in both factories (Harare 76.2%, Bulawayo 84.3%) and multi-cutter operations in Bulawayo (81.3%) had the highest proportion of airborne concentrations above the OEL. Overall, 60.3% and 58.6% of measurements in the Harare and Bulawayo factories, respectively, exceeded the OEL.

Figure 1 and Figure 2 show changes in the mean personal exposure chrysotile asbestos fibre concentrations at the Harare and Bulawayo factories, respectively, from 1996 to 2016. Annual personal mean exposure levels generally showed a downward trend over the years, with high levels recorded from 1996 to 2001 for the Harare factory, and from 1996 to 2007 for both factories, in almost all operational areas. Personal exposure concentrations below the OEL began after 2008, except for fettling table operations in the Harare factory and multi-cutter operations in the Bulawayo factory. For the Harare factory, cutting saw operations, ground hard waste operations, laundry rooms, and kollergang operations exhibited high levels of personal exposure chrysotile fibre concentrations in or before 2007, with a considerable decline thereafter. The Bulawayo factory also showed a similar trend in all operational areas in which high personal exposure concentrations were observed in or before 2008.

Table 4 and Table 5 show the mean personal exposure chrysotile fibre concentrations by time- period for the Harare and Bulawayo factories, respectively.

For the Harare factory, the overall percentage decline over the time periods was 14.3% from the time period 1996–2000 to 2001–2008, and a 50% decline was registered from the period 2001–2008 to 2009–2016. The Bulawayo factory showed a generally similar pattern to that of the Harare factory, with an overall factory exposure decline of 21.4% between the periods 1996–2000 and 2001–2008, while a 45.5% decline in personal exposure chrysotile fibre concentrations was registered between the time periods 2001–2008 and 2009–2016. Overall, during the period 1996–2000, exposure levels ranged from 0.11–0.18 f/mL, compared to 0.04–0.12 f/mL personal exposure chrysotile fibre concentrations recorded for the period 2009–2016 for the Harare factory. Similarly, for the Bulawayo factory, personal exposure chrysotile fibre concentrations ranged from 0.09–0.22 f/mL during the earlier years of 1996–2000, compared to 0.03–0.10 f/mL recorded for the period 2009–2016.

### Observations

Observations made during site visits and during data gathering at the factories noted that manufacturing equipment and ventilation systems were generally in good condition. Respiratory protective equipment such as dust masks was provided, and cleaning of floors and other operations was carried out under wet conditions. These observations were made to check the state of the manufacturing equipment, and whether good work practices were being followed.

Furthermore, personnel in the factories indicated that the equipment in use has been in operation since the 1990s and the 2000s, and that such equipment was subject to regular maintenance.

Essentially, personal sampling during the period 1996–2016 took place—and still takes place—in areas where raw chrysotile fibres or asbestos products are processed and handled.

Raw fibre is supplied in plastic-wrapped bags and stored on site. The bags are moved to the fibre preparation (or fibre treatment) area, where they are loaded into fibre preparation machines called kollergang. The fibre is manually tipped into the kollergang machine following opening of the bags with a knife. This area provides much scope for fibre release in the workplace when bags are opened and tipped into the kollergang machines and, hence, may explain the rather elevated personal exposure fibre concentrations observed at the kollergang operational area. However, some form of ventilation is provided that produces a positive or inward draft into the kollergang machines at the fibre entry point.

A slurry of fibre cement is processed through a continuous flow process to form chrysotile asbestos cement sheets, and this is performed through a controlled computerized system while an operator is in the control cabin. The corrugated chrysotile cement sheets are then lifted from the production line by an automated machine, stacked on pallets, and taken by forklifts to areas such as the sawing/cutting areas.

Other ancillary operations include the sawing or cutting of chrysotile cement sheets and facia boards to size using powered saws equipped with local exhaust ventilation. The moulded goods section, where various goods are moulded under wet conditions, is generally labour intensive. Furthermore, in Bulawayo, lathe machines are used to cut asbestos pipes, prepare joints and couplings, and polish products. Discussions with personnel at the two factories indicated that workers had always been—and were still being—provided with masks, and that they were monitored to check whether they followed the good work practices set by the factories.

It was also noted that the ACM factories have followed international best practices in manufacturing, occupational safety and health, and environmental management systems throughout the period 1996–2016. This has led to ACM factories being accredited to ISO 9001, ISO14001, and OSHAS 18001 (now ISO 45001). Such accreditations suggests that the ACM factories endeavour to provide a safe work environment. The factories have been certified to the international standards indicated above since 2001. The period 1996–2000 was characterized by a build-up towards certification as reported by the company personnel in the ACM entities; hence, because of pursuing such standards, this may have contributed towards the downward trends in chrysotile asbestos fibre concentrations over the years. While there was a general decline in industrial production across the country over the period 2001–2008—and, in particular, an accelerated decline from 2006 to 2008, due to hyperinflation—the good occupational safety and health framework may also have been a contributing factor to the downward trend as the ACM factories strived for continuous improvement in their business processes.

## 4. Discussion

This study constitutes the single largest personal exposure chrysotile asbestos fibre concentration dataset in Zimbabwe. The general decline in exposure over time from 1996 to 2016 suggests good occupational safety and health (OSH) framework implementation by the two factories over the years, with the years after 2008 showing much lower concentration levels below the OEL. Decreasing trends in personal exposure chrysotile asbestos fibre concentrations may also be viewed from the perspective that industry was responding to the anticipated lowering of the airborne chrysotile fibre OEL as a result of increased calls to ban all forms of asbestos, triggering the scaling up of exposure controls in the factories. However, at cutting saw operational areas, personal exposure chrysotile fibre concentration levels suggest high-risk activity in both factories during the earlier years of 1996–2008, perhaps due to weak controls, as fibre concentrations considerably exceeded the OEL.

During the period 1996–2000, economic activity was generally high, and this may have contributed to the high personal exposure chrysotile asbestos fibre concentrations observed during these early years. Economic instability, however, set in during the period 2001–2008, which resulted in a significant decline in industrial production across all industries, which may also have contributed to decreases in personal exposure chrysotile asbestos fibre concentrations. Although production rates were not available from the asbestos cement manufacturing factories, it is widely known that production across all industries—including the ACM factories—in Zimbabwe was seriously affected by hyperinflation during the period 2001–2008, such that in 2008 there was almost an economic standstill situation in the country, which could thus have contributed to the observed decline in personal exposure chrysotile asbestos fibre concentrations. Zilaout et al. (2020), in a study on trends in respirable dust and respirable quartz concentrations in the European industrial minerals sector over a 15-year period, cited macroeconomic developments as affecting trends, and postulated that recession may have contributed to the downward trends observed [25]. From 2009 to 2016, there was general stability in the economy, with most companies back to optimal operation. Retooling of operations and systems was made easier as the country adopted a multicurrency system, with the US dollar being the main currency of use. This may have contributed to improved OSH programmes which, in turn, could also possibly have contributed to further decline in personal exposure chrysotile asbestos fibre concentrations during this period.

Despite the overall decline in occupational personal exposure chrysotile asbestos fibre concentrations based on an occupational exposure limit of 0.1 f/mL, descriptive statistics for both factories suggest that there was overexposure among those exposed—especially during the period 1996–2000. The Harare factory shows that 60.3% of personal exposure chrysotile asbestos fibre concentrations exceeded the OEL, while for the Bulawayo factory, 58.6% of personal exposure chrysotile asbestos fibre concentrations exceeded the OEL (Table 2 and Table 3, respectively). The exposure limit of 0.2 f/mL adopted by the chrysotile asbestos industry in Zimbabwe was consistent with the threshold limit value (TLV) set by the American Conference of Governmental Industrial Hygienists (ACGIH), which adopted a TLV of 0.2 f/mL during the early-to-mid-1990s. During the 1990s, mean personal exposure chrysotile asbestos fibre concentrations at various operational areas did not exceed this TLV and, as such, the industry was of the view that they were generally making the necessary efforts to prevent overexposure. However, with new information and knowledge on the risk profile for chrysotile asbestos exposure since the mid-1990s to date, personal exposure chrysotile asbestos fibre concentrations could be considered as presenting a risk of disease if evaluated against the OEL of 0.1 f/mL.

Although observations made during site visits showed that the general state of the factories was good, the Bulawayo factory appeared to show better housekeeping than the Harare factory, and it may be assumed that the Bulawayo factory may have been maintaining such good housekeeping, thus possibly contributing to fewer chrysotile asbestos fibre concentrations above the OEL compared to the Harare factory. Additionally, the Harare factory had always had a greater proportion of workers over the years compared to the Bulawayo factory, suggesting more activity and handling of chrysotile asbestos and associated products, which could thus have also contributed to more personal chrysotile asbestos fibre concentrations above the OEL compared to the Bulawayo factory.

The overall downward trend in personal exposure fibre concentrations observed over the years 1996–2016 is consistent with patterns observed in other places where chrysotile asbestos cement products were being produced [26,27]. In a study where 2089 asbestos exposure datasets were put together from 1995 to 2006, asbestos exposure levels were shown to decrease from 0.92 f/mL in 1996 to 0.60 f/mL in 1997, to 0.19 f/mL in 1998, and to 0.06 f/mL in 1999, and this decrease was considered as possibly being due to enforcement of legislation and the banning of the use of amosite and crocidolite. The mean asbestos fibre concentration in the asbestos cement plants was recorded as 0.31 f/mL [25], whereas in the Zimbabwe factories, the overall mean personal exposure chrysotile asbestos fibre concentration was 0.11 f/mL. However, specific exposure patterns in the chrysotile asbestos cement plants worldwide became limited after 2000, as most countries banned the use, handling, and production of chrysotile asbestos [20]. Nevertheless, in Germany, there was a steady decline in asbestos exposure between 1950 and 1990 in textile, cement brake pads, and drilling/sawing operations [1,2,9].

In this study, personal exposure chrysotile asbestos fibre concentrations in the chrysotile cement asbestos pipe manufacturing industry ranged from 0.03 to 0.30 f/mL. In Thailand, breathing zone asbestos concentrations in cement pipe production ranged from 0.12–2.13 f/mL between 1987 and 1988 [2,28]. Thus, the declining pattern in personal exposure asbestos fibre concentration estimates over the years in Thailand is similar to the declines in concentrations observed in this study. Creely et al., in a review on trends in inhalation exposure, also reported decreases in respirable fibre levels in various workplace settings involving possible exposure to asbestos, where asbestos fibre concentrations were reported to decline by as much as 32% per annum. Regulatory intervention, good occupational hygiene practices, and improved ventilation were factors cited as contributing to decreasing temporal trends [10].

In another study by Albin et al. in a Swedish asbestos cement factory, the authors reported asbestos fibre concentrations declining from 1.5–6.3 f/mL in 1956 to 0.3–5 f/mL in 1969 and 0.5–1.7 f/mL in 1975 [13]. Higashi et al. evaluated personal exposure at two Japanese manufacturing and processing plants producing asbestos-containing products such as roofing sheets, and reported that asbestos fibre concentrations ranged from 0.05 to 0.78 f/mL [15]. In this study, overall, during the earlier years of 1996–2000, personal exposure chrysotile asbestos fibre concentrations ranged from 0.11 to 0.18 f/mL and 0.09 to 0.22 f/mL in the Harare and Bulawayo factories, respectively. Additionally, declines in asbestos fibre concentrations over the years in various workplace settings have been reported [29,30]. These decreasing trends in asbestos fibre concentrations over time, as observed and reported in the various studies mentioned above, are also consistent with the declines observed in personal exposure chrysotile fibre concentrations in the chrysotile asbestos cement manufacturing industry in Zimbabwe.

### Strengths and Limitations of the Study

The considerable large amount of airborne chrysotile fibre concentration data collected over a long period of time—spanning two decades—using recognized standard asbestos methods and equipment, and being unique in Zimbabwe, offers a key point of strength to this study. Additionally, a collection of such chrysotile exposure data could be used as a basis for future epidemiological studies. However, within each factory, there were years for which measurements were not available in various operational areas in each calendar year, resulting in fibre concentration data gaps in some years and, thus, affecting fibre concentration patterns over the years. Furthermore, production rates—which may have provided further insights in personal exposure fibre concentration patterns for the period 1996–2016—were not available from the factories. However, notwithstanding these limitations, the considerable amount of data provides insights into changes in personal exposure chrysotile asbestos fibre concentrations over time, serving as input data for future research.

## 5. Conclusions

The personal exposure chrysotile asbestos fibre measurements collected over two decades—from 1996 to 2016—in key operational areas of the factories aided in a comprehensive analysis of trends in personal exposure chrysotile asbestos fibre concentrations in the asbestos cement manufacturing industry in Zimbabwe. Personal exposure chrysotile fibre concentration data in the two factories show a downward trend over the years, with high concentrations being exhibited in or before 2008. These findings are consistent with the downwards trends over time observed in other studies. The Harare factory showed more overexposure than the Bulawayo factory. Wet processes should continue to be applied in order to continuously sustain reductions in levels of exposure to airborne chrysotile asbestos fibres in the workplace. These results can aid in future epidemiological studies, serve as a basis for the establishment of a job-exposure matrix for similar workplace settings, and assist in predicting the possible occurrence of asbestos-related diseases in Zimbabwe.

## Figures and Tables

**Figure 1 ijerph-18-10755-f001:**
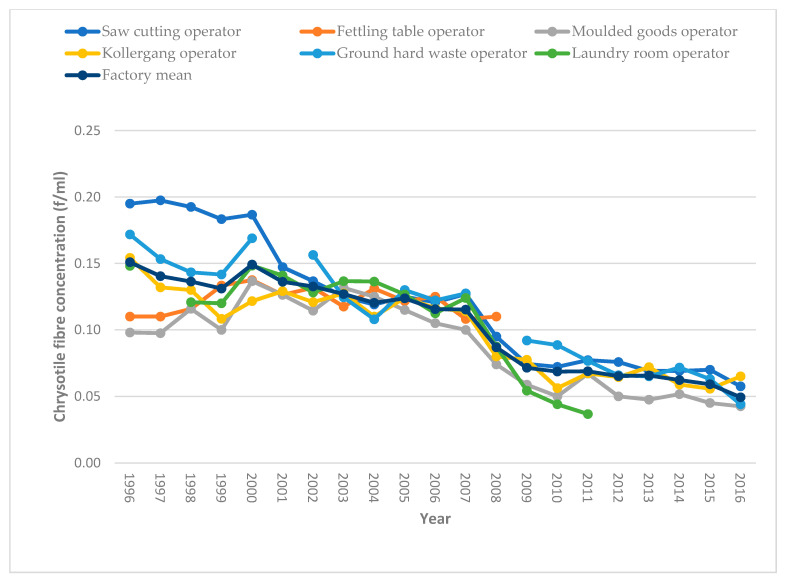
Changes in personal exposure chrysotile asbestos fibre concentrations from 1996 to 2016 for the Harare factory.

**Figure 2 ijerph-18-10755-f002:**
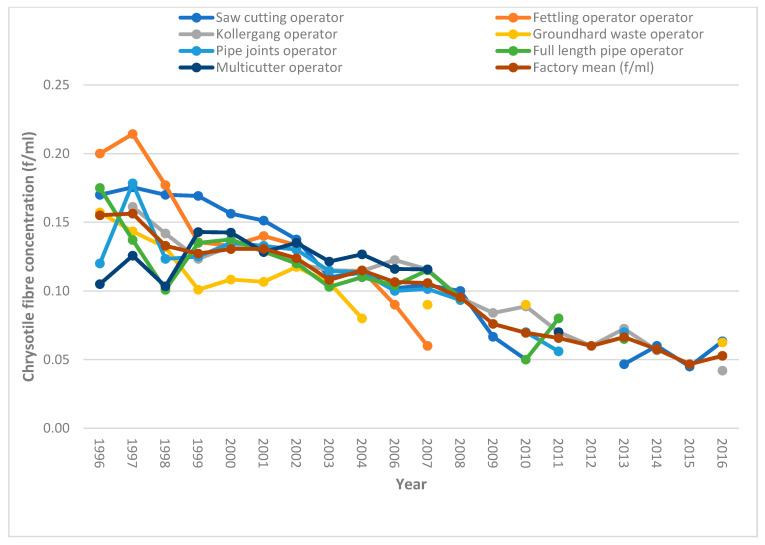
Changes in personal exposure chrysotile asbestos fibre concentrations from 1996 to 2016 for the Bulawayo factory.

**Table 1 ijerph-18-10755-t001:** Description of tasks and operational areas used in analysing personal chrysotile fibre measurements in the Harare and Bulawayo factories.

Task/Operational Area	Description of Task
Saw Cutting operations	Operator cuts chrysotile asbestos cement sheets and facia boards to size.
Fettling table operations	Scrapping of unwanted chrysotile asbestos cement matter on finished moulded goods such as ridges, garden ware, and polishing using sandpaper by operators.
Kollergang operations	Operator opens ~50 kg chrysotile asbestos bags using a knife and loads the fibre into the process machine.
Moulded goods table operations	Operators mould various goods under wet conditions, such as ridges, or garden ware goods such as flower vessels.
Ground hard waste operations	Operator feeds chrysotile asbestos cement waste material into grinder machine for recycling back into process.
Pipe section—lathe machining of chrysotile asbestos pipe joints	Operators operate lathe machines such as sewer lathe, Lang lathe, broad bend lathe, and Geminis lathe machines by machining joints so that they are ready for coupling pipes.
Pipe section—lathe machining of full-length chrysotile asbestos sewer/water pipes.	Operator operates lathe machines—namely, Faben, Voith, and O&S lathe machines—to prepare full-length pipe for a joint, and polish joint with sandpaper.
Pipe section—multi-cutter operations	Cutting full-length pipes into collars used for coupling pipes using a multi-cutter machine.

These tasks were considered to have potential for highest exposure to airborne chrysotile asbestos.

**Table 2 ijerph-18-10755-t002:** Mean airborne personal chrysotile asbestos fibre concentrations (f/mL) for various operations/tasks in the Harare chrysotile cement manufacturing factory.

Job/Task	N	Mean	SD	Range		% >0.1 f/mL *
				Min	Max	
Cutting saw operator	(225)	0.12	0.05	0.03	0.24	60.9
Fettling table operator	(126)	0.12	0.02	0.05	0.19	76.2
Moulded goods operator	(192)	0.10	0.04	0.03	0.20	46.4
Kollergang operator	(203)	0.10	0.04	0.04	0.20	54.2
Ground hard waste operator	(168)	0.12	0.04	0.02	0.22	63.7
Laundry room operator	(149)	0.12	0.04	0.03	0.21	73.8
Overall factory	(1063)	0.11	0.04	0.04	0.18	60.3

N: actual number of personal samples, 1996–2016; *: % measurements greater than OEL of 0.1 f/mL.

**Table 3 ijerph-18-10755-t003:** Mean airborne personal chrysotile fibre concentrations (f/mL) for various operations/tasks in the Bulawayo chrysotile cement manufacturing factory.

Job/Task	N	Mean	SD	Range		% >0.1 f/mL *
				Min	Max	
Cutting saw operator	(113)	0.13	0.04	0.01	0.24	75.2
Fettling table operator	(51)	0.16	0.06	0.06	0.30	84.3
Kollergang operator	(111)	0.11	0.04	0.03	0.24	60.4
Ground hard waste operator	(64)	0.12	0.04	0.04	0.24	54.7
Pipe joints operator	(99)	0.12	0.03	0.04	0.30	69.7
Full length pipe operator	(97)	0.12	0.03	0.04	0.30	67.0
Multi-cutter operator	(64)	0.12	0.03	0.05	0.20	81.3
Overall factory	(600)	0.11	0.04	0.03	0.22	58.6

N: actual number of personal chrysotile asbestos samples, 1996–2016; *: % measurements greater than OEL of 0.1 f/mL.

**Table 4 ijerph-18-10755-t004:** Mean personal exposure chrysotile asbestos fibre concentrations (f/mL) by period between 1996 and 2016: Harare factory.

Job/Task	Time Period	N	Mean	SD	95% CI		Range	
					LB	UB	Min	Max
Saw cutting operator	1996–2000	60	0.19	0.01	0.19	0.19	0.16	0.24
	2001–2008	88	0.13	0.02	0.12	0.13	0.08	0.18
	2009–2016	77	0.07	0.02	0.07	0.08	0.03	0.11
Fettling table operator	1996–2000	53	0.12	0.04	0.11	0.13	0.05	0.19
	2001–2008	73	0.12	0.02	0.12	0.13	0.04	0.20
	2009–2016	nil						
Moulded goods operator	1996–2000	58	0.11	0.04	0.10	0.12	0.04	0.20
	2001–2008	82	0.11	0.04	0.11	0.12	0.03	0.18
	2009–2016	52	0.05	0.01	0.05	0.06	0.03	0.08
Kollergang operator	1996–2000	58	0.13	0.03	0.12	0.14	0.05	0.20
	2001–2008	81	0.12	0.02	0.11	0.12	0.04	0.16
	2009–2016	64	0.07	0.02	0.06	0.07	0.04	0.11
Ground hard waste operator	1996–2000	57	0.16	0.03	0.15	0.16	0.08	0.22
	2001–2008	56	0.13	0.03	0.14	0.13	0.03	0.20
	2009–2016	55	0.07	0.02	0.07	0.08	0.02	0.17
Laundry room operator	1996–2000	47	0.13	0.03	0.12	0.14	0.06	0.20
	2001–2008	87	0.13	0.02	0.12	0.13	0.06	0.21
	2009–2016	15	0.05	0.01	0.04	0.05	0.03	0.07
Overall factory	1996–2000	60	0.14	0.02	0.14	0.15	0.11	0.18
	2001–2008	92	0.12	0.02	0.12	0.12	0.07	0.18
	2009–2016	80	0.06	0.01	0.06	0.07	0.04	0.12

N: number of personal chrysotile asbestos fibre samples; SD: standard deviation; 95% CI: 95% confidence interval; LB: lower bound; UB: upper bound; Min: minimum; Max: maximum.

**Table 5 ijerph-18-10755-t005:** Mean personal exposure chrysotile asbestos fibre concentrations (f/mL) by period between 1996 and 2016: Bulawayo factory.

Job/Task	Time Period	N	Mean	SD	95% CI		Range	
					LB	UB	Min	Max
Cutting saw operator	1996–2000	50	0.17	0.02	0.16	0.18	0.12	0.24
	2001–2008	49	0.12	0.02	0.11	0.12	0.09	0.16
	2009–2016	14	0.06	0.02	0.05	0.07	0.01	0.08
Fettling table operator	1996–2000	40	0.17	0.06	0.16	0.19	0.07	0.30
	2001–2008	11	0.12	0.03	0.10	0.14	0.06	0.15
	2009–2016							
Kollergang	1996–2000	36	0.14	0.03	0.13	0.15	0.08	0.24
	2001–2008	42	0.12	0.01	0.11	0.12	0.08	0.14
	2009–2016	33	0.07	0.03	0.06	0.08	0.03	0.18
Ground hard waste	1996–2000	44	0.13	0.04	0.11	0.14	0.07	0.24
	2001–2008	15	0.11	0.04	0.10	0.11	0.08	0.13
	2009–2016	5	0.07	0.02	0.05	0.09	0.04	0.09
Pipe joints	1996–2000	44	0.13	0.04	0.12	0.14	0.06	0.30
	2001–2008	46	0.11	0.01	0.11	0.12	0.08	0.15
	2009–2016	9	0.06	0.01	0.05	0.07	0.04	0.08
Full length pipe operator	1996–2000	43	0.13	0.04	0.12	0.14	0.06	0.27
	2001–2008	45	0.11	0.01	0.11	0.11	0.07	0.14
	2009–2016	9	0.07	0.02	0.05	0.08	0.04	0.09
Multi-cutter operator	1996–2000	26	0.13	0.04	0.11	0.14	0.05	0.20
	2001–2008	36	0.12	0.01	0.12	0.13	0.10	0.14
	2009–2016	2	0.07	0.03	0.02	0.32	0.05	0.20
Overall factory	1996–2000	51	0.14	0.03	0.13	0.15	0.09	0.22
	2001–2008	50	0.11	0.01	0.12	0.11	0.09	0.15
	2009–2016	45	0.02	0.01	0.06	0.07	0.03	0.10

N: number of personal chrysotile asbestos fibre samples; SD: standard deviation; 95% CI: 95% confidence interval; LB: lower bound; UB: upper bound; Min: minimum; Max: maximum.

## Data Availability

The dataset used in this study are available from the corresponding author on reasonable request. The datasets are not publicly available to maintain confidentiality of the factories used in the study.

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
