# Peer review of "Trends in Airborne Chrysotile Asbestos Fibre Concentrations in Asbestos Cement Manufacturing Factories in Zimbabwe from 1996 to 2016"

_ijerph, 2021, doi:10.3390/ijerph182010755_

Round 1

Reviewer 1 Report

The present study provides some information to know the workplace concentration of asbestos fibers in the Zimbabwean chrysotile asbestos-cement manufacturing industry from 1996 to 2016. However, there are considerable problems to be solved.

Firstly, the level of asbestos fiber concentration in the workplace reported is not reflecting the actual exposure level of workers because they were using respiratory protectors. How correctly the workers used the protecting equipment is not known, thus actual exposure levels are unknown. So, this paper is just reporting the asbestos fiber concentration levels in the workplace air and the present title which indicates actual exposure of workers is not adequate.

     Secondly, it is clear from the figures 1 and 2 that the asbestos fiber concentrations in the workplaces have decreased during the observation period with some exceptional cases. It is not necessary to conduct ANOVA nor linear/multiple regression analysis to show this trend in the asbestos concentration. Comparison between time-period and years in not important. Also, the statistical analysis is not clarifying actual factors such as equipment in the workplace, amount of asbestos used, handling way, etc. that contributed to lowering the asbestos concentration. These factors should have affected the air-born asbestos fibers but no information is provided if there were any changes in these factors across time-period or years. As the statistical analysis cannot contribute to clarify this point, it would be useless and thus should be omitted.  Further, the authors used the linear point imputation method to deal with missing data, however, it is not adequate to apply such method because there was so much amount of missing data.

     Consequently, I would like to recommend the authors to omit the statistical analysis, and instead just to present original data by figures or tables. Missing data should be excluded. Data on the measurements greater than OEL (%) are important thus it would be meaningful to present it by, for example, a figure plotted by observation years. Finally, the process by which the original data was provided to the author by the factory (that it was formally received by the factory rather than personally provided by the person in charge) must be clearly explained.

Title  (Trends in Occupational Exposure to Chrysotile Asbestos Fibre in Asbestos Cement Manufacturing Factories)

⇒ Trends in Workplace Chrysotile Asbestos Fiber Concentration in Asbestos Cement Manufacturing Factories

L214  However, data analysis was performed based on total number of personal exposure levels after the linear point imputation method in SPSS, to deal with missing data.

⇒ Many of the values obtained by the interpolation method are used in place of missing values, and a significant portion of the reported data are estimates rather than measured values. There is no explanation as to why the missing values occurred. The analysis should be done using only the measured values without using the interpolation. At Bulawayo factory, the measured value is less than half.  In addition, there seems to be a mistake in the value of overall factory in the table 3. The missing value is described as Nil in the Supplementary Information. Does it mean “not identifiable level”?  If this means levels below the lower limit of detection, the lower limit of 0.02 f / ml should be used for analysis.

L220 Analysis of variance (ANOVA) was used to assess differences in mean personal exposure in the time periods 1996 – 2000, 2001 – 2008 and 2009 – 2016. A Tukey Post Hoc Test (Tukey’s Honest Significance Difference test) was run to find out which specific group means of time periods (compared with each other) were different.

⇒ In investigating the change of exposure concentration depending on the time-periods, ANOVA was conducted in three periods of 1996-2000, 2001-2008, 2009-2016, but the objective basis of the standard for each period is not sufficiently explained. The explanation of L154-166 is not enough. Such analysis would not identify any specific factors that contributed to the improvement of fiber concentration.

L336  3.2. Linear and Multiple Regression Analysis

⇒ As for model 2 regression analysis, there is insufficient explanation on what were used as the independent variables. It is presumed that the calendar year and time-period were used, however both are indicators of time change and thus it is considered inappropriate to use both even if multicollinearity is taken into consideration. Figures 1 and 2 would be sufficient to show the change (decrease) of the exposure concentration over time in the past. Since there is no point in making future predictions, it is less meaningful to clarify more appropriate predictive variables in these statistical analyzes, and regression analysis is unnecessary.

L232  Logistic regression analysis was used to predict whether cases (in this case whether exposure levels above the OEL) of exposure levels were correctly classified as being above or below the OEL, from the independent variables of year and time-period.

⇒ It is stated that a logistic regression analysis was performed to see which of the year and time-period was better as a predictor of the percentage of exposure above OEL, but the scientific and technical implications are small. Neither variable is a variable that cannot be used for future prediction or management. Also, it is meaningless to compare the odds ratio between the time-period, which is the exposure period divided into three, and the year, which is the 21 division, when the exposure level is decreasing during the same period.

L291 while in the Bulawayo factory, fettling table operations, though exhibiting high exposure levels before 2003, showed a major decline to non-detectable levels after 2011.

⇒ The results of the fettling table operations at the Bulawayo plant are not shown in Fig. 2. The figure is incomplete as it does not match the ANOVA table. Furthermore, both Fig1 and Fig2 contain line segments that are not explained in the caption.

L384 3.4. Observations

⇒ When and by whom were these observations made? Probably from factory records, but without these details, the value of the data is very low. In addition, it is described as respiratory protective equipment was provided. It is very important if the workers used the respiratory protectors and how correctly they used such protectors. Anyway, if the workers have used such equipment, the exposure concentration described in this study would be the workplace concentration and not be reflecting the personal exposure levels. If so, it is necessary to make an overall correction of the description of the manuscript including the title of the paper.

L412-421 

⇒ This is just a result, not a discussion, should be described in the result section.

L442  Additionally, Harare factory had always had more workers over the years compared to the Bulawayo factory and thus could also have contributed to more exposure levels above the OEL compared to the Bulawayo factory.

⇒ It is logically strange to raise the difference in the number of workers as the cause of the many observations exceeding OEL.

Author Response

Thank you very much for the comments on our manuscript. We have benefitted from your insights into the study. Our responses are as per the attachment. 

Reviewer 2 Report

The paper reported the trends in occupational exposure to chrysotile asbestos fibre in asbestos cement manufacturing factories in Zimbabwe, 1996 to 2016. The results can be used in future epidemiological studies and in predicting the occurrence of asbestos related diseases in Zimbabwe.

The paper was concise and well organized and deserves to be published. There are some suggestions, which would improve the quality of the paper but are not essential for publication.

  1. The poor discussion of the results. Author just shows the great amount of results that they have achieved, but they did not use them to develop an interesting discussion which could supplement to earlier studies onepidemiological studies。
  2. In the section of Discussion, why the author did not compare asbestosrelated diseaseswith that of other epidemiological studies. In my opion,
  3. References: Many of the references have been superceded and more modern ones are required.
  4. Please correct the reference “J. Toxicol Environ Health 2007a, 70, 1076-1107”. The authors should check the references carefully. The format should be consistent.

Author Response

Thank you so much for the comments which helped us to think through of  a possible epidemiology study which can arise from the data set used. Attached is our responses to your comments.

Round 2

Reviewer 1 Report

Several amendments have been made but the present paper still has some problems, most of which are I have pointed out in my previous review.

Data source:

⇒ The authors wrote “The data were extracted from the paper records of personal chrysotile asbestos fibre exposure measurements done in the factories by company personnel. (L127)” and “The authors would like to thank the chrysotile asbestos cement manufacturing factories for availing (providing? having made available?) their chrysotile asbestos exposure data which spanned about two decades without which the study could not have materialized(L524)”. However, it must be clearly mentioned in the text (methods section) that the original data (records of exposure measurements) were provided from the company to the authors on official agreements between the factories and the authors. In the cover letter from the authors, they wrote “Following a request by BM to undertake the study in the ACM factories permission was granted and access to the factories was given after which extraction of the personal chrysotile asbestos fibre concentration data from paper records of the chrysotile asbestos measurements was done with the assistance of factories personnel.” Further, information on the methods explained in the text (L131-155) are, I suppose, written based on the information from the factory personnel or the paper records. These points should be clearly mentioned because measurements have not been done by the authors. Current description (L127: The data were extracted from the paper records of personal chrysotile asbestos fibre exposure measurements done in the factories by company personnel.) is not enough.

ANOVAs:

⇒ The observation period is divided into three unequal periods without any rational background or criteria, and the difference in concentration between these three periods is examined by analysis of variance. This analysis is meaningless because the criteria and grounds for setting the three periods are unclear. The decreasing tendency of the concentration throughout the whole period is clear from the figures and tables, and ANOVA is to be omitted as suggested previously. In Park et al [25], the time span of 1995 to 2006 was divided into 4 equal length time-periods so that ANOVA is justified as a method to examine the difference or changes along with time.

Title (Trends in Occupational Personal Chrysotile Asbestos Fibre Concentrations in Asbestos Cement Manufacturing Factories in Zimbabwe, 1996 to 2016)

⇒ The title has been amended but not adequately. “Personal Chrysotile Asbestos Fibre Concentrations” is not understandable. If the authors want to emphasize that they have used personal sampling methods, then I would suggest “Trend in Airborne Chrysotile Asbestos Fibre Concentrations measured with personal samplers at Asbestos Cement Manufacturing Factories in Zimbabwe from 1996 to 2016” or so on.

“personal chrysotile asbestos fibre concentrations”:

⇒ This phrase is used so many times, but as with the title, its meaning is unclear. It should be amended. I would suggest “ambient chrysotile asbestos fibre concentration of workers” or so on. However, I also think "personal exposure concentrations" could be used if it is clearly explained in the first part of the paper that it means not the actual exposure level of workers (because they were protected by using adequate respirators) but the workers ambient concentration.

Supplementary Materials:

⇒ Results of the linear and multiple regression analysis and logistics regression analysis are still presented as the supplementary materials. It has been pointed out that the applications of these analysis have problems and they have been omitted in the paper. Thus, they should not be presented as the supplementary materials. Description from L504-517 should also be amended accordingly.

Author Response

Thank you for taking time to review our manuscript. Attached is our responses.

Reviewer 2 Report

The authors have considered all comments raised by the reviewers and revised the manuscript accordingly based on these comments. The revision is fine and can be accepted for publication in current form.

Author Response

Thank you for taking time to review our manuscript.
